# Transcriptome Analysis of Flower Development and Mining of Genes Related to Flowering Time in Tomato (*Solanum lycopersicum*)

**DOI:** 10.3390/ijms22158128

**Published:** 2021-07-29

**Authors:** Hexuan Wang, Yahui Yang, Yiyao Zhang, Tingting Zhao, Jingbin Jiang, Jingfu Li, Xiangyang Xu, Huanhuan Yang

**Affiliations:** Laboratory of Genetic Breeding in Tomato, Key Laboratory of Biology and Genetic Improvement of Horticultural Crops (Northeast Region), Ministry of Agriculture and Rural Affairs, College of Horticulture and Landscape Architecture, Northeast Agricultural University, Harbin 150030, China; whxxx36@126.com (H.W.); yyh201296@126.com (Y.Y.); zyy19976409@126.com (Y.Z.); 15504506671@sohu.com (T.Z.); jjb1248@126.com (J.J.); lijf_2005@126.com (J.L.)

**Keywords:** tomato, RNA-seq, flowering time, genetic pathways

## Abstract

Flowering is a morphogenetic process in which angiosperms shift from vegetative growth to reproductive growth. Flowering time has a strong influence on fruit growth, which is closely related to productivity. Therefore, research on crop flowering time is particularly important. To better understand the flowering period of the tomato, we performed transcriptome sequencing of early flower buds and flowers during the extension period in the later-flowering “Moneymaker” material and the earlier-flowering “20965” homozygous inbred line, and we analyzed the obtained data. At least 43.92 million clean reads were obtained from 12 datasets, and the similarity with the tomato internal reference genome was 92.86–94.57%. Based on gene expression and background annotations, 49 candidate genes related to flowering time and flower development were initially screened, among which the greatest number belong to the photoperiod pathway. According to the expression pattern of candidate genes, the cause of early flowering of “20965” is predicted. The modes of action of the differentially expressed genes were classified, and the results show that they are closely related to hormone regulation and participated in a variety of life activities in crops. The candidate genes we screened and the analysis of their expression patterns provide a basis for future functional verification, helping to explore the molecular mechanism of tomato flowering time more comprehensively.

## 1. Introduction

Flowering is a crucial developmental process of angiosperms. It marks the transformation from vegetative growth to reproductive growth [1,2]. The reproductive ability of plant offspring, viability of the population, and adaptability to the ecological environment are closely related to flowering. Flowering time of crops has a profound effect on fruit yield and quality; therefore, adaptive flowering time is an important agronomic trait that has received a great deal of attention from breeders [3,4]. Flowering time is regulated by internal and external factors [5,6,7]. The vegetative growth of crops must be accumulated to a certain extent prior to flowering. Crops can adapt to changes in environmental conditions such as seasonal changes, light, humidity, and temperature and will accept corresponding flowering instructions. Flower development can be divided into four stages according to time: (1) the flowering transition period; (2) meristem determination of gene activation; (3) activation of genes determining flower organ characteristics; (4) flower organ formation. Among these stages, many genes related to flowering time have been identified in particular in the flowering transition stage [8,9].

Previous studies have identified some key regulatory genes, their functions in inducing flowering, and the associated molecular mechanisms in *A. thaliana*. Thus far, it has been determined that there are six genetic pathways involved in the regulation of flowering time, which are related to the photoperiod, vernalization, temperature, gibberellin, autonomy, and age [10,11,12]. In the photoperiod pathway, after the photoreceptors of plants receive light signals, they rely on the biological clock to initiate or inhibit flowering [13]. It is well established that the part of the plant that acts as a photoperiod receptor is the leaf and that long-day periods promote flowering. CONSTANS (*CO*) and FLOWERING LOCUS T (*FT*) are the main regulatory genes in the photoperiod pathway [14,15,16]. *CO* is a transcription factor with a zinc finger structure, and light promotes its accumulation, thus upregulating *FT* expression. *FT* is a flower-forming hormone that acts as a flowering signal. It is synthesized in leaves and transferred to the stem tip, where it induces accelerated flowering [17,18,19]. The process whereby low temperature induces plants to flower is referred to as vernalization. In *A. thaliana*, the core gene of the vernalization pathway is FLOWERING LOCUS C (*FLC*). *FLC* inhibits the expression of *FT* to prevent early flowering [20,21]. *FLC* also acts on autonomic pathways. As a common plant hormone, gibberellin promotes flowering by inducing the expression of FT. It can also positively regulate flowering through the interaction of the DELLA protein with the age-dependent transcription factor *SPL9* [22,23].

Unlike *A. thaliana*, tomatoes are day- and night-neutral plants. In the process of regulating flowering time, the developmental stage and genotype play dominant roles. In tomato, SINGLE FLOWER TRUESS (*SFT*) is a homologous gene of *FT* from *A. thaliana*, which is a key gene controlling the flowering transition and meristem specificity. Under both short- and long-day conditions, *SFT* mutants show delayed flowering, sepals transform into leaves and carpels, and other floral organs are reduced in size [24]. SELF PRUNING (*SP*) is a homologous gene of CENTRORADIALIS and *TFL1*. *SP* was initially detected in the stem tip and leaves and was later found in the middle leaves of the inflorescence. *SP* was shown to delay tomato flowering. Mutation of *SP*-induced premature flowering altered the sympodial nodes of the biaxial branches and caused a change from infinite to finite growth with a pattern of two inflorescence cappings, resulting in shortening [25,26]. *SFT* and *SP* have similar functions in the inflorescence structure and floral meristem, and they interact with each other in the network that regulates flowering time [27]. Uniflora (*UF*) is also one of the key genes that controls the flowering time of tomatoes. The *uf* mutant is susceptible to seasonal and light intensity conditions, especially in winter, and it shows postponed flowering and no inflorescences, which are replaced by unisexual flowers [28]. Compound inflorescences (*S*) and jointlessness (*J*) are both regulated downstream of *UF* and jointly participate in the regulation of flowering time. The *s* mutant produces compound inflorescences and shows slowed flower formation progress, while the *j* mutant resumes growth after reaching the 1–3-flower stage and exhibits delayed flowering. The *s*:*uf* and *j*:*uf* double mutants bloom later than any of the single mutants [29,30]. Both *SP3D* and *SP5G* are homologous to *A. thaliana FT*. *SFT*/*SP3D* actively initiates flower opening, while *SP5G* has the opposite effect. However, under long exposure to sunshine, *SP5G* plays a positive role, and plants with mutations in this gene exhibit early flowering [31,32]. *tmf* is an early-flowering mutant with single-flowered inflorescences and large sepals with leaf-like shapes. *TMF* has also been found to be a gene that negatively regulates tomato flowering. AN was activated prematurely in the tmf mutant, leading to early flowering phenotype [33].

Transcriptome sequencing technology can produce accurate transcript information for specific tissue research materials within a specific period and is an effective method for studying differentially expressed genes (DEGs) and analyzing candidate genes. We chose “Moneymaker” and “20965”, two materials with different flowering periods (the flowering period of “20965” is earlier than that of Moneymaker), and the experimental time points were set in the bud germination period and the flower development period respectively. We used the RNA-seq sequencing platform to analyze DEGs and mine and screen candidate genes related to flowering time. The purpose was to provide theoretical support for studying the molecular mechanism of tomato flowering regulation, the specific positions of candidate genes, and directional breeding for flowering traits.

## 2. Results

### 2.1. Statistics and Quality Verification of Transcriptome Sequencing Data

To obtain transcriptome data for the two tomato materials with different flowering times, we completed transcriptome sequencing during the early flower bud formation and flower maturity stages (Table 1 and Appendix A). The reference genome was the NCBI SL3.0 assembly. Three biological replicates were performed for each species and each period to obtain twelve sample data points. The average amount of clean data per sample was found to be 6.64 g when the project was run on the DNBSeq platform. In second-generation sequencing, a corresponding quality value is given for each base measured. This quality value is a measure of the accuracy of sequencing, where Q20 represents an error rate of 1% and Q30 represents an error rate of 0.1%. As shown in Table 1, after the data were filtered, more than 97% of the reads showed a quality value ≥ Q20, and the Q30 base distribution percentage was between 92.66% and 94.34%. In this experiment, at least 43.92 million clean reads were obtained. Thereafter, we used HISAT to align the clean reads with the tomato reference genome, and the comparison efficiency was 92.86–94.57%. More than 90.6% of the clean reads were located in unique positions in the reference genome. The Pearson correlation coefficients of all gene expression values between each pair of samples were calculated, and the correlation coefficients of the 12 datasets (23,627 genes in total) are presented in the form of heat maps (Figure 1). These graphs reflect the correlations of gene expression between samples—the higher the correlation coefficient is, the more similar the gene expression level is.

### 2.2. Differential Gene Expression Analysis

In the transcriptomes obtained in different tomato varieties and different developmental stages of inflorescences, we set the following criteria for DEG screening: log2-fold change ≥1 and Q value ≤0.05 (Table 2 and Figure 2). In MM_1-vs-MM_2, in total of 5980 DEGs were detected, among which 3152 genes were upregulated and 2828 genes downregulated. The largest number of DEGs was identified in the 20965_1-vs-20965_2 group, at 8779, with 4954 genes showing upregulated expression and 3825 genes downregulated expression. For MM_1-vs-20965_1, there were 79 DEGs, of which 214 were upregulated and 565 downregulated. In the MM_2-vs-20965_2 comparison group, 1345 DEGs were identified, including 859 upregulated genes and 486 downregulated genes. The results show that the DEGs that regulate flowering were more highly expressed during petal expansion. The Venn diagram of the DEGs shows the numbers of DEGs identified in each comparison group and the overlap relationships between the comparison groups. In total, 40 genes appeared in more than one period and variety, and these genes may be closely related to the regulation of flowering time (Figure 3).

### 2.3. GO and KEGG Enrichment Analyses of DEGs

The DEGs screened in the four comparison groups MM_1-vs-MM_2, 20965_1-vs-20965_2, MM_1-vs-20965_1, and MM_2-vs-20965_2 were subjected to GO and KEGG enrichment analyses. The Gene Ontology (GO) database is a comprehensive database of descriptions of gene functions. Through GO enrichment analysis, identified DEGs can be functionally annotated. GO terms are divided into three main categories: molecular functions, biological processes, and cellular components. DEGs were enriched in 49 functional branches, including 28 biological processes, 18 cell components, and 13 molecular functions (Appendix A). In the biological process category, the terms that were significantly enriched were “cell process”, “metabolic process”, “response to stimulation”, and “biological regulation”. Among them, the DEGs enriched in the “translation” pathway were the most abundant in the term “cellular process”. In total, 282 DEGs enriched in the “response to stimulus” term were closely related to the defense response function (Figure 4A and Appendix A). In the cell composition category, “cell”, “cell component”, and “organelle” were terms related to tomato flowering. In this type of GO term, the description of the functional pathway most enriched in DEGs was plasma membrane, which belongs to the term “cell” (Figure 4B and Appendix A). In the molecular function category, DEGs were obviously enriched in two branches, “binding” and “catalytic activity”. Among them, 1110 DEGs were enriched in the “ATP binding” pathway of the term “binding” (Figure 4C and Appendix A). GO classification indicated that the flowering-related DEGs of tomato were mainly related to cell metabolism, binding, and catalytic processes.

The Kyoto Encyclopedia of Genes and Genomes (KEGG) database contains information on intracellular molecular interaction networks, bioinformatics findings, and specific biological changes. Through KEGG pathway analysis, insight into the biological functions of genes can be gained. As shown in Figure 5, the size of the bubble represents the number of genes annotated to the KEGG pathway. Generally, functions with a Q value ≤ 0.05 are considered significantly enriched. In the MM_1-vs-MM_2 group, genes were significantly enriched in the “photosynthesis” (k00195), “metabolic pathway” (k01100), “biosynthesis of secondary metabolites” (k01110), “carbon metabolism” (k01200), “plant hormone signal transduction” (k04075), and “circadian rhythm-plant” (k04712) pathways, among other physiological pathways (Figure 5A and Appendix A). In the comparison of 20965_1-vs-20965_2, the greatest accumulation of DEGs was found in pathways such as “metabolic pathway” (k01100), “plant hormone signal transduction” (k04075), “MAPK signal transduction pathway-plant” (k04016), “carbon metabolism” (k01200), “secondary metabolites biosynthesis” (k01110), and “starch and sucrose metabolism” (k00500) (Figure 5B and Appendix A). In the comparison of different flower development periods in the two materials, “metabolic pathway” (k01100), “biosynthesis of secondary metabolites” (k01110), “phenylpropane biosynthesis” (k00940), “amino acid biosynthesis” (k01230)), “amino sugar and nucleotide sugar metabolism” (k00520), and “circadian rhythm-plant” (k04712) were the pathways showing the greatest enrichment of DEGs (Figure 5C and Appendix A). The pathways enriched by DEGs were similar in several comparison groups, focusing on metabolism, phytohormones, photosynthesis, and circadian rhythm. These pathways are jointly involved in regulating flower development in tomatoes, resulting in different flowering periods in different materials.

### 2.4. Analysis of Genes Related to Tomato Flowering Time and Flower Development

We jointly screened 59 candidate genes related to flowering time and flower development from the two flower development stages of the late-flowering Moneymaker variety and the early-flowering “20965” variety based on gene expression, Log2 fold change, KEGG Pathway Description, GO_C Desc, GO_F Desc, GO_P Desc, GeneBank Desc, and combined with related references (Table 3 and Appendix A). Most of the DEGs were related to the photoperiod pathway, of which three *COP1*-related genes and three *CONSTANS*-related genes were upregulated. Moreover, their expression levels were upregulated more significantly in the “20965” variety. The expression of cryptochromes *CRY2* was increased. Flowering locus t (*FT*), GIGANTEA (*GI*), and WNK5 protein expression was asio upregulated. Two early-flowering single genes (*ELF3*, and *ELF4*) were upregulated, and PELPK1 protein expression was increased. BEE-, bHLH-, and MYB-related transcription factors were also closely related to the photoperiod pathway and showed upregulated expression. When crops undergo the photoperiod pathway to induce flowering, they also need proper circadian rhythms. *COP1*, cryptochrome, early-flowering genes, and GIGANTEA (*GI*) were all enriched in the circadian rhythm pathway. Two genes with the MADS-box structure were thought to be involved in the vernalization response. There were 11 genes related to the gibberellin pathway, including two *GA2* oxidases (*GA2ox2* and *GA2ox4*), a *GA20* oxidase (*20ox-2*), two FRUITFULL genes (*FUL1* and *FUL2*), the GA receptor DELLA protein GIBBERELLIC ACID INSENSITIVE1 (GAI), two GA receptors GIBBERELLIN INSENSITIVE DWARF1 (GID1ac and GID1b-1), an MYB transcription factor family gene, an *SOC* gene rich in the MADS-box structure, and one histidine kinase. Their expression levels were all upregulated. There were two genes in the age pathway, of which only *AP2**-like* was enriched for negative regulation of the cytokinin-activated signaling pathway, showing downregulated expression, and the expression of AP2a gene increased. The transcription factor *PIF4* was upregulated in the temperature pathway, while ACTIN-related protein 6 (ARP6) and ACTIN-related protein 7 (ARP7) and the short vegetative growth period gene SHORT VEGETATIVE PHASE (*SVP*) were downregulated in the temperature pathway. Furthermore, it has a negative effect on flowering. In the autonomous pathway, both single flower truss (*SFT*) and flowering promoting factor protein 3 were upregulated, while FRIGIDA (*FRI*) expression was downregulated. The other related candidate genes were not classified into a certain pathway, but they also regulated flowering and flower development. PRK2, PRK1, and LOC101243937 can promote pollen germination and regulate pollen tube growth, and their expression levels during flower development were higher than those at the beginning of flower buddin. A gene described as zeatin O-glucosyltransferase affects the vegetative growth and tissue growth of crops, thereby regulating flower morphogenesis. The expression of BLADE-ON-PETIOLE genes (*BOP2* and *BOP3*) was downregulated. The remaining genes such as *TAA1* were all upregulated to promote flower development. The *JAZ2* gene acts on the jasmonic acid regulatory pathway to accelerate flowering. Some candidate genes that promote flowering were more upregulated in “20965” cultivars than in Moneymaker cultivars, and all candidate genes that inhibit flowering were more downregulated in “20965”. This result shows that the dynamic changes and behavior patterns of these genes may be the key reason for the “20965” flowering period earlier than Moneymaker.

### 2.5. WGCNA of DEGs

Weighted gene coexpression network analysis (WGCNA) is a tool suitable for various complex data analyses. Gene clustering and module division were performed using the dynamic shearing algorithm, and finally 14 gene co-expression modules were obtained (Figure 6). The number of genes in the module ranges from 29 (yellow-green) to 4153 (brown). Each branch of the clustering tree represents a module. Each leaf represents a gene. Each color represents a module. Observing the correlation coefficients of the modules, it was found that the genes in the green-yellow and orange modules had higher specificity during the flowering period of “20965” varieties. KEGG pathway analysis was performed on the green-yellow module, and it was found that genes were enriched in “phenylalanine metabolism”, “phenylpropanoid biosynthesis”, “plant hormone signal transduction” and other pathways (Appendix A). The genes in the orange module are on multiple metabolic pathways such as “galactose metabolism”, “tyrosine metabolism”, and “starch and sucrose metabolism” (Appendix A). GA2ox2 candidate genes are located in this module.

### 2.6. DEGs Bioregulation Analysis

Using MapMan software, the biological regulation and metabolism-related roles of the DEGs were analyzed in the early flower bud and flower extension stages of the two tomato cultivars (Figure 7). The results show that the DEGs mainly responded to indole acetic acid (IAA), abscisic acid (ABA), brassinosteroids (BR), ethylene, cytokinins, jasmonates (JA), salicylic acid (SA), and gibberellin (GA) hormone regulation. The DEG expression results of the samples of the two varieties in the two periods were similar. There were also upregulated and downregulated genes associated with each hormone. There were more upregulated genes than downregulated ones. The DEGs involved in the IAA pathway were the most abundant, and the DEGs involved in the jasmonate pathway were the least abundant. Regarding redox reactions, the DEGs participated in biological reactions regulated by calcium regulation, receptor kinases, G-proteins, MAP kinases, phosphoinositides, carbon, nutrients, and light. In addition, several DEGs accumulated in a variety of oxidoreductase categories, such as the heme, thioredoxin, ascorb/glutathione, glutaredoxin, and dismutase/catalase categories. In addition, many DEGs were enriched in the transcription factor, protein modification, and protein degradation categories. Regarding the regulation of cell responses, most DEGs involved in cell division and the cell cycle showed downregulated expression, and the upregulated expression of DEGs involved in developmental processes increased significantly. In the biotic and abiotic stress response pathways, there were more upregulated genes than downregulated ones. The results of the analysis of transcription factors among the DEGs identified in MM_1-vs-MM_2 and 20965_1-vs-20965_2 are shown in the figure. These transcription factors mainly included members of the AP2-EREBP, bHLH, G2-like, WRKY, bZIP, MADS, MYB, MYB-related, NAC, C2H2, HB, histone, putative DNA-binding, chromatin remodeling, and other families, and most were upregulated. These families play important roles in regulating auxin signaling, ethylene signaling, meristems, chromosome structures, etc. The expression of most WRKY-family genes increased during the flowering period, whereas the expression of DEGs related to histone and chromatin remodeling decreased. This may provide proof that the flowering of crops is transitioning from vegetative growth to reproductive growth at this time.

### 2.7. Validation of RNA-Seq Data by qRT-PCR

We selected eight DEGs for qRT-PCR analysis to verify the accuracy of the RNA-seq results. These 8 DEGs were randomly selected from the 49 candidate genes identified as potentially playing roles in flowering time regulation. Real-time fluorescent quantitative PCR with the gene-specific primers designed in the NCBI database (Appendix A) was used to analyze flower samples at different developmental stages to quantify relative expression. It could be seen from the figure that under the background that *p* < 0.05 represents a significant difference, the data marked ‘a’ was significantly higher than ‘b’ and ‘c’; ‘b’ was significantly higher than ‘c’. There was no significant difference between the same letter. The results show that the relative expression levels of the samples determined by qRT-PCR were basically the same as the sample FPKM values measured by RNA-seq (Figure 8), thus proving that the RNA-seq data are reproducible and reliable.

## 3. Discussion

The second-generation molecular sequencing technology RNA-seq has become the most common method used for studying gene expression and screening and predicting candidate genes. Flowering is an important part of crop growth and development. It is a key sign of the transition of crops from vegetative growth to reproductive growth. The completion of this transformation at an appropriate time is necessary for crops to survive. Therefore, transcriptome sequencing was conducted in two flower development stages of two tomato varieties with different flowering periods in the present study, and the transcriptome data were verified by qRT-PCR. We designed two time points, flower bud period and flower development period, in order to obtain candidate genes related to early flowering and flower development. The results prove that the data obtained by RNA-seq are reliable. The flowering period of “20965” is earlier than that of Moneymaker. Flower bud period and flower exhibition period were the two time points we set up. Based on the analysis of transcriptome data, many DEGs were identified in the four selected comparison groups. According to the DEG analysis, there is a larger difference between the developmental stages within a genotype than between genotypes at the same developmental stage. The genes could respond to different flowering times in different varieties. That is, the amount of gene expression changed according to different flower development stages. In GO enrichment analysis, the “cell” and “cell component” categories were enriched with the most genes. In addition, they were significantly enriched in the “response to stimulation”, “binding”, and “catalytic activity” categories, and these terms were all related to the flower development of crops. In the KEGG pathway enrichment analysis, we found that, although the selected comparison groups were different, the most DEGs were enriched in the pathways of metabolism, phytohormones, photosynthesis, and compound synthesis, which was similar to the results of previous studies. This shows that the above pathways are closely related to the regulation of crop flowering.

To increase the reproduction rate and provide a good developmental basis for fruit setting, crops need to bloom at the right time; therefore, crops must undergo a series of morphological and physiological changes under the common influence of the external environment and endogenous factors. Factors such as temperature or light can be regarded as external signals. Substances such as hormones or different compounds form an interaction network with each other to induce crops to produce flower buds from stem-end meristems and to flower at the right time. *A. thaliana* is a model crop for studying flowering time of crops. Scholars have explored six pathways that regulate flowering: the photoperiod, vernalization, autonomy, temperature, age, and gibberellin pathways [34,35,36]. Alternation between light and darkness is an important factor that affects the transformation of flowers. A crop will perceive changes in the length of light exposure (i.e., the photoperiod) through its own internal clock [37]. PHYA, PHYB, CRY, etc. are photoreceptors through which crops perceive light signals. Depending on the light intensity, the life activities of crops change regularly in a 24 h cycle, forming a circadian rhythm [38]. The CIRCADIAN CLOCK ASSOCIATED1 (*CCA1*) and LATE ELONGATED HYPOCOTYL (*LHY*) genes in the MYB transcription factor family are located upstream in the circadian network and negatively regulate the expression of TIMING OF CAB EXPRESSION 1 (*TOC1*) (Figure 9). *PRR9*, *PRR7*, *PRR5*, and *TOC1* are members of the same family. Their activity peaks during the day, and, together with *LHY* and *CCA1,* they are responsible for the morning cycle [39,40,41]. The red/far-red photoreceptor PHY and the blue photoreceptor CRY play key roles in achieving complete consistency of the internal cycle of crop plants with environmental signals and accurately controlling flowering time. *CRY* promotes flowering. A *CRY* unigene (*CRY2*) was selected in this study, and the expression level was found to increase with the development of flowers, indicating that *CRY* is positively correlated with flowering [42]. *COP1* is the central repressor of photomorphogenesis in higher plants and acts as a RING E3 ubiquitin ligase downstream of the photoreceptor and upstream of the *GI* gene [43,44]. *COP1* interacts with GI, while GI and FKF1 form a complex to promote the expression of *CO*. Thereafter, the downstream gene *FT*, which is regulated by *CO*, receives instructions to induce crops to flower. In this transcriptome, COP1 and its homologs and GI-like genes were detected, and their expression was found to be upregulated. The early flowering genes *ELF3* and *ELF4* are rhythmically expressed in long-day crops. Their expression reaches a peak near dusk, and both gene products promote flowering. Woe-Yeon Kim and others constructed a ZTL OX elf3-1 double mutant of *A. thaliana* and found that the rhythm was disrupted and that flowering was delayed [45]. It was concluded that *ELF3* does not rely on *CO* and *FT* to function. *ELF3**-like* and *ELF4-like* genes were all present in the transcriptome, their expression patterns were similar, and they were more active during the flowering period. It is speculated that early flowering genes may regulate flowering time throughout the entire flowering period. The temporal and spatial regulation of *CO* makes it a key transcription factor in the photoperiod-dependent flowering induction pathway. The N-terminal domain of *CO* consists of two consecutive zinc finger domains, B1 and B2, and the C-terminal domain consists of a CCT domain that can bind to DNA to form a complex that activates the transcriptional regulation of downstream genes. CO-like proteins play a role similar to that of *CO* in regulating the flowering of crops. In a study of winter flowering in sugar beet, Nadine et al. found that two CO-like proteins lacking different domains showed epistatic interactions and jointly promoted sugar beet blooming [46]. The key *CO* gene was found in the transcriptome, and similar proteins (CO1 and CO3) were also identified. The identified GO functions mainly involved genes that regulate flower development and biological rhythms, and the expression of *CO* and similar genes was observed in flowers. The maximum accumulation observed during flower expansion indicates that *CO* positively regulates flowering, and proteins similar to *CO* have similar functions. The *PELPK* motif consists of 36 unique pentapeptide repeats. In *A. thaliana*, the PELPK1 strain shows slower growth and delayed flowering relative to the wild type. However, lines overexpressing this gene show early flowering, proving that *PELPK1* is a positive regulator that promotes growth and flowering [47]. In this experiment, PELPK1-like was enriched in the pathway that regulates photoperiod and flowering (GO: 2000028); therefore, in tomato, PELPK1 and similar proteins are expected to actively regulate flowering.

The autonomous pathway is another important route whereby flowering is regulated in tomatoes, and the key genes in this pathway are *SFT*, *J*, and *FRI*. FRI promotes the methylation of histone H3K4 (H3K4me3) in *FLC*, which in turn upregulates the expression of FLC, activates a MADS-box transcription factor, and inhibits the flowering of crops [48,49]. In this study, only one gene encoding FRIGIDA-like protein 3 (101252492) was selected, and no *FLC*-related genes were obtained. The expression level of 101252492 in the initial stage of flower bud differentiation was higher than that in the flower development stage. It is speculated that *FRI* functions in the meristem at the initial stage of flowering to activate downstream gene expression and inhibit early flowering. In previous reports, the BLADE-ON-PETIOLE protein has mostly been found to function in leaves and lateral organs. In a study of the tomato BOP protein, Xu found that BOP1, BOP2, and BOP3 all interact with *TMF* in the nucleus to inhibit flowering and cause complications in inflorescences [50]. In *A. thaliana*, BOP1 and BOP2 also delay the flowering of *A. thaliana* by downregulating *FD* in the shoot meristem [51]. The 101245709 and 10124600 genes were shown to encode BOP3 and BOP2 proteins. Their expression changes were consistent with previous conclusions. These genes were actively expressed in the transitional stage in meristems, and their expression then decreased in the mature stage of flowers. The function of double BOP3 and BOP2 mutants can be verified to explore whether SlBOPs affect other aspects of crop growth and development and to study the function of this protein family more comprehensively. In the zeatin biosynthesis pathway, we identified two upregulated zeatin O-glucosyltransferase genes. GO analysis revealed that genes were enriched in the cytokinin metabolism process (GO: 0009690), leaf morphogenesis (GO: 0009965), root morphogenesis (GO: 0010015), regulation of vegetative meristem growth (GO: 0010083), flower morphogenesis (GO: 0048439), and other pathways. Studies have shown that *OscZOG1* is preferentially expressed in meristems and new organs, induces lateral root development, and improves the agronomic traits of rice [52]. Therefore, it is reasonable to speculate that 101247407 promotes flower morphogenesis, but the specific regulatory mechanism is still unclear, and follow-up work is needed.

Gibberellin plays a role in regulating organ growth and promoting reproduction and development throughout the growth period of crops. Under both long-day and short-day, chrysanthemums need gibberellin to induce flowering [53]. The biosynthesis of GA occurs through the action of two subfamily members, *GA20ox* and *GA30ox*. The DELLA proteins are growth inhibitors that constitute a subfamily of GRAS proteins. They may bind to transcription factors because they contain a conserved GRAS domain. GA degrades DELLA by activating the expression of GID1 and prevents DELLA proteins from exerting their growth-inhibitory effect [54,55]. In this transcriptome sequencing study, the identified candidate genes in the gibberellin pathway were all highly expressed during the blooming stage. Their interactions and synergistic activities with other pathways help gibberellin maintain a dynamic balance in crops. In turn, gibberellin promotes flower development and regulates the formation of flower organs. Other plant hormones also play roles in the complex network that regulates the flowering of crops and positively or negatively affects the flowering period. Jasmonic acid signals are generally modulated by biotic and abiotic stresses in crops, but studies have shown that excessive *JAZ2* can promote tomato leaf growth and early flowering [56]. In chrysanthemums, excessive *JAZ1-like* levels cause bud formation and late flower development to occur at a significantly later time in transgenic plants than in the wild type [57]. This shows that the same hormone pathway may be affected by crosstalk between the regulation of the crop response to stress and the regulation of crop growth and development, resulting in the same type of genes performing multiple functions in crops.

Transcription factors can bind downstream genes to induce their expression and play important roles in growth and development, metabolism, and the responses of plants to the external environment. According to our MapMan analysis, AP2-EREBP, bHLH, G2-like, WRKY, bZIP, MADS, MYB, MYB-related, and other transcription factor families play roles in flowering time regulation in Moneymaker and “20965” homozygous inbred line. The bHLH family is involved in the formation of plant morphology, and the *SPATULA* gene is a member of the bHLH family, whose activity has been detected in pollen tissues. The flower morphology of *A. thaliana* mutants with *SPATULA* functional loss was established previously [58]. bHLH transcription factors also promote the synthesis of crop secondary products under stimulation by jasmonic acid signals. According to KEGG pathway analysis, the DEGs identified in this experiment were enriched in the synthesis pathways of various compounds, such as “secondary metabolite biosynthesis” (k01110) and “phenylpropane biosynthesis” (k00940). It is speculated that members of the bHLH transcription factor family can induce the production of secondary products. The synthesis and expression of related genes have an impact on flowering. The 101252729 and 101265165 genes encode bHLH transcription factors, and their expression levels showed an upward trend during the flowering stage. According to our GO analysis, these genes respond to blue light. They are predicted to promote flowering. MADS-box transcription factors play a regulatory role during the entire growth period of crops, regulating not only flowering time, flower development, and floral organ formation but also fruit ripening and hormone signal transduction [59]. The *AGL*, *FUL*, and *SOC* genes related to flowering time all promote the occurrence of flowering, and they are all involved in the catabolism of gibberellin. Members of the MYB transcription factor family contain a conserved MYB domain that has been shown to be widely involved in metabolic regulation, stress resistance, and hormone transduction in crops. Jasmonic acid-ZIM domain proteins interact with the R2R3-MYB transcription factors MYB21 and MYB24 and thereby affect jasmonic acid to regulate the development of stamens in *A. thaliana*. Jasmonic acid-ZIM domain proteins also upregulate the expression of *MYB21* through interactions to promote flower opening in tomatoes [60]. In this study, the expression of *MYB21* in the inbred lines and wild-type materials was very abundant during the flowering period and played an important role in tomato flower development.

In conclusion, this study constructed comprehensive tomato transcriptomes from Moneymaker material and the “20965” homozygous inbred line and screened 49 candidate genes related to tomato flowering time and flower development. By considering six different flowering pathways, we analyzed and predicted the functions of these genes. Their expression levels in the two different varieties from the flower bud stage to the flower development stage showed the same trend, that is, the expression of genes that promote flowering was upregulated, and the expression of genes that inhibited flowering was decreased. Moreover, the expression levels of most candidate genes in the “20965” homozygous inbred line were greater than those in the Moneymaker line. The behavioral changes of these genes may be the reason that the “20965” blooms earlier than Moneymaker. On the basis of previous studies, we screened out other genes that may regulate flowering and flower development, providing a reference for a more systematic and comprehensive study of tomato flower development networks.

## 4. Materials and Methods

### 4.1. Experimental Material

The research materials were selected from the experimental base of the Tomato Research Group at the Horticulture Station of the College of Horticulture and Landscape Architecture, Northeast Agricultural University. The two varieties were Moneymaker (identified as “MM” in the transcriptomic analysis, where “MM” corresponds to “Control” in the raw sequencing data) and the homozygous inbred line “20965” (represented by “20965” in the transcriptomic analysis, where “20965” corresponds to “Treat” in the raw sequencing data). The homozygous inbred line “20965” blooms earlier than Moneymaker. Two plants showing good growth and no diseases or pests were selected from each material. Flower buds were collected and immediately put into liquid nitrogen after being picked and were then stored in a refrigerator at −80 °C for later use.

### 4.2. RNA-Seq Library Construction and Sequencing

Total RNA (Thermo Fisher, Waltham, MA, USA) was extracted from three biological replicates in each group in the early stage of flower bud development and flower development according to the kit requirements. The purity and integrity of the extracted RNA were tested by gel electrophoresis, and the concentration of RNA at 260 and 280 nm was detected. The extracted total RNA was enriched with polyA mRNA by dT magnetic beads. After adding the fragmentation buffer, the mRNAs were fragmented. The first cDNA strand was synthesized using random primers and reverse transcriptase, and the second cDNA strand was then synthesized using DNA polymerase I and RNase H. The cDNA fragments were purified. We used mRNA fragments as templates to reverse transcribe cDNA. After the library was constructed, the appropriate fragments were selected for PCR amplification. Sequencing was performed by the BGI Tech platform (Shenzhen, China), the library was generated with the MGISEQ-200 (Shenzhen, China), and the read length was 150 bp.

### 4.3. Sequencing Reads and DEG Analysis

The data obtained by sequencing are referred to as raw reads and include some reads with adapters and low-quality reads. These sequences can interfere with subsequent analysis and reduce the credibility of the data. Therefore, it is necessary to perform filtering to obtain clean reads. SOAPnuke is filtering software independently developed by BGI. It can remove linker contamination, unknown N base contents greater than 5%, and low-quality reads (defined according to the ratio of bases with a quality value of less than 10 among the total number of bases in the reads, where reads with a ratio greater than 20% are considered low-quality reads) [61].

Sequence annotation information was obtained by alignment of the filtered sequences with the GCF_000188115.4_SL3.0 reference genome of *S. lycopersicum* by Hisat software [62]. Gene expression levels were calculated based on the gene length and sequencing depth, and reads were mapped to the genome and normalized to the number of reads per million from each kilobase length of a gene [63]. Based on the Poisson distribution, DEGs in the transcriptome were identified by the DEGseq method. In this study, genes with a log2|fold change (FC)| ≥ 1 and false discovery rate ≤0.05 were defined as differentially expressed. The Venn diagram was drawn using R language.

### 4.4. GO and KEGG Analysis

Gene Ontology (GO) analysis can comprehensively define the attributes of genes and gene products in an organism, mainly because GO provides a dynamic standard vocabulary with which to describe these attributes. A Q value (corrected *p* value) ≤ 0.05 was defined as the threshold, and DEGs that met this condition were defined as significantly enriched. The life activities of crops are not carried out independently but work together under the mutual influences of different genes and cooperative regulation. Therefore, to explore the complex functions of genes, pathway analysis was carried out. The KEGG pathway database is the main public database focused on pathways. Sequences were aligned to the KEGG database, significant enrichment analysis was performed, the hypergeometric test was applied, and the pathways with Q values ≤0.05 were defined as showing significant enrichment in DEGs. KEGG analysis can determine the most important metabolic and signal transduction pathways involving the identified genes [64].

### 4.5. Weighted Gene Coexpression Network Analysis (WGCNA)

The WGCNA v1.48 package was used to construct a gene coexpression network. In this study, the expression levels of 12 transcriptome samples (2 time points of 2 materials, 3 repetitions each) were selected for WGCNA analysis. For missing values, if the gene expression of a sample was missing more than 50%, it was eliminated; if a gene was missing in more than 50% of the samples, it was eliminated. For the filtering of gene expression, the top 75% of the genes with the median absolute deviation (MAD) were retained. The filtered data used the automatic network construction function blockwiseModules to construct a co-expression network. The filtered data constructed a co-expression network through the automatic network construction function blockwiseModules. The network type is unsigned, and the correlation type is Pearson. In the module, the exceptions were that the power was 9, the TOM similarity threshold was 0.19, and the minimum number of genes was 20. In order to study the highly correlated modules, the Pearson correlation coefficient between the sample matrix and the gene module was calculated and statistically tested. The larger is the correlation coefficient, the higher is the correlation between the module and the sample. We used Pearson’s correlation coefficient >0.7 as a specific module for further analysis. 

### 4.6. Analysis of MapMan Biological Functions of DEGs

MapMan is pathway analysis software designed specifically for plants, in which most pathways are arranged manually. Based on the CDS of the tomato genome, the mapping file was constructed using the annotation website (https://www.plabipd.de/query_view.ep, accessed on 29 March 2021). The expression levels of DEGs in the comparison groups of MM_1-vs-MM_2 (MM_1-vs-MM_2 corresponds to Control1-vs-Control3 in the raw sequencing data), 20965_1-vs-20965_2 (20965_1-vs-20965_2 corresponds to Treat1-vs-Treat3 in the raw sequencing data), MM_1-vs-20965_1 (MM_1-vs-20965_1 corresponds to Control 1-vs-Treat1 in the raw sequencing data), and MM_2-vs-20965_2 (MM_2-vs-20965_2 corresponds to Control 3-vs-Treat3 in the raw sequencing data) were uploaded. Then, we chose PageMan functional analysis with MapMan software to annotate the biological functions of the differentially expressed genes between each sample.

### 4.7. Real-Time Quantitative PCR Analysis

Real-time quantitative PCR was used to test the reliability of the RNA-seq data. In this experiment, eight DEGs were randomly selected. Primers were designed based on the NCBI database, and reverse transcription and fluorescence measurements were performed according to the requirements of the kit (Vazyme, Nanjing, China). Three technical replicates were performed for each biological replicate of each sample. Relative quantitative data were calculated according to the ΔΔCT method: normalization (ΔCT = CT (sample) − CT(GAPDH)); ΔΔCT = ΔCT (sample A) − ΔCT (sample B); relative quantification = 2^−ΔΔCT^. The qRT-PCR data were analyzed with SPSS v.21.0 software for variance analysis, and the Waller–Duncan (W) method was used for comparison at the *p* < 0.05 level.

## Figures and Tables

**Figure 1 ijms-22-08128-f001:**
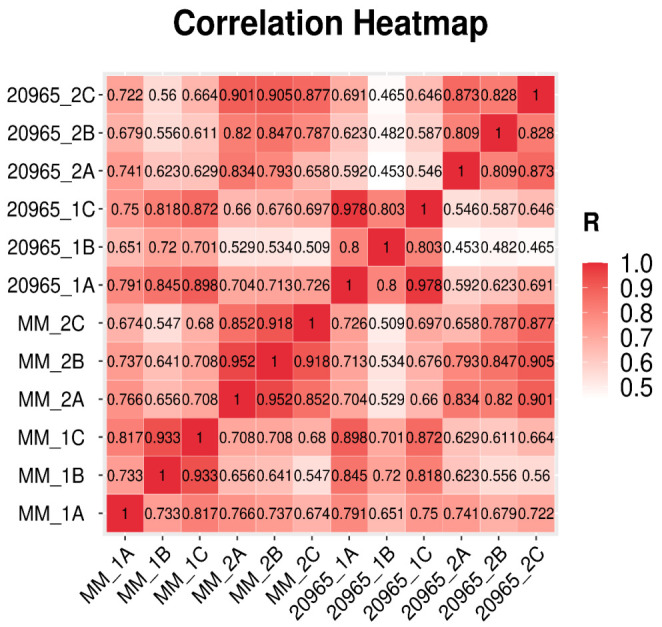
Pearson correlation coefficients of all 12 samples. The Pearson correlation coefficients of all gene expression levels between every two schemes.

**Figure 2 ijms-22-08128-f002:**
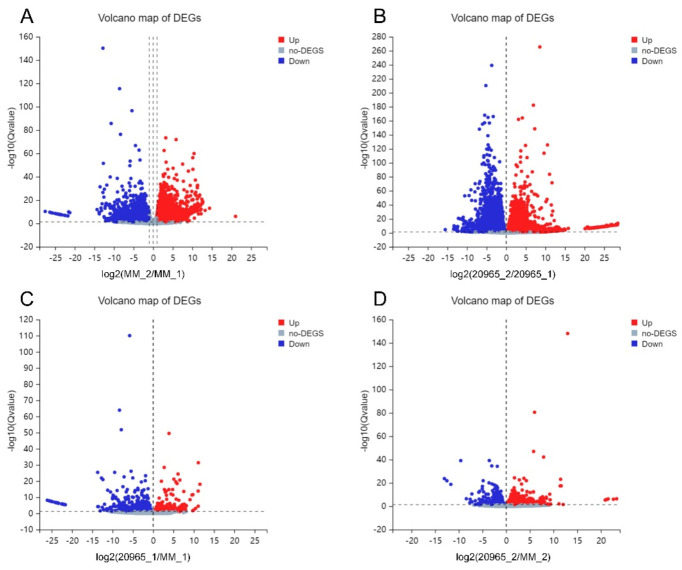
Volcano map of DEGs. (**A**) Volcano map of MM_1-vs-MM_2. (**B**) Volcano map of 20965_1-vs-20965_2. (**C**) Volcano map of MM_1-vs-20965_1. (**D**) Volcano map of MM_2-vs-20965_2. The red dots represent upregulated genes, the blue dots represent downregulated genes, and the gray dots represent non-DEGs. The X-axis represents the fold change of the difference after conversion to log2, while the Y-axis represents the significance value after conversion to -log10.

**Figure 3 ijms-22-08128-f003:**
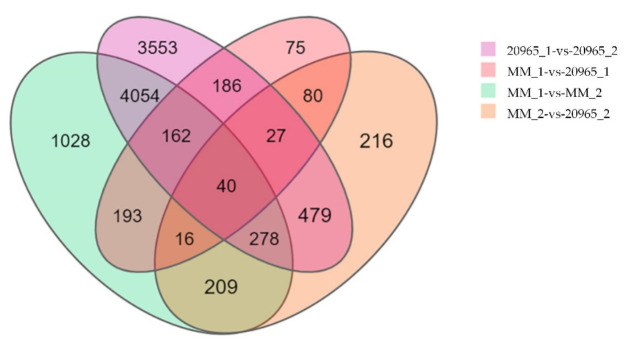
Venn diagram of DEGs included in the four groups. The four groups are MM_1-vs-MM_2, 20965_1-vs-20965_2, MM_1-vs-20965_1, and MM_2-vs-20965_2. Each circle represents a group of gene sets. The overlapping area of different circles represents the intersection of these gene sets, that is, the co-expressed genes. Non-overlapping parts indicate the uniquely expressed genes. The numbers on the figure represent the number of DEGs in the corresponding area.

**Figure 4 ijms-22-08128-f004:**
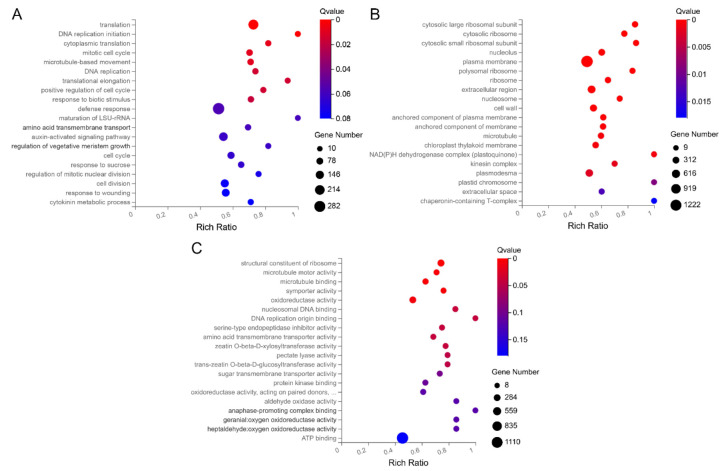
GO functional enrichment of differentially expressed genes in tomato flower buds. (**A**) GO_biological process enrichment bubble chart. (**B**) GO_cellular component enrichment bubble chart. (**C**) GO_molecular function enrichment bubble chart. The sizes of the bubbles indicate the number of genes enriched in the GO term. The color of the bubble represents the Qvalue. The rich factor is the ratio of differentially expressed gene numbers annotated in this pathway term to all gene numbers annotated in this pathway term. The greater the rich factor is, the greater the degree of pathway enrichment.

**Figure 5 ijms-22-08128-f005:**
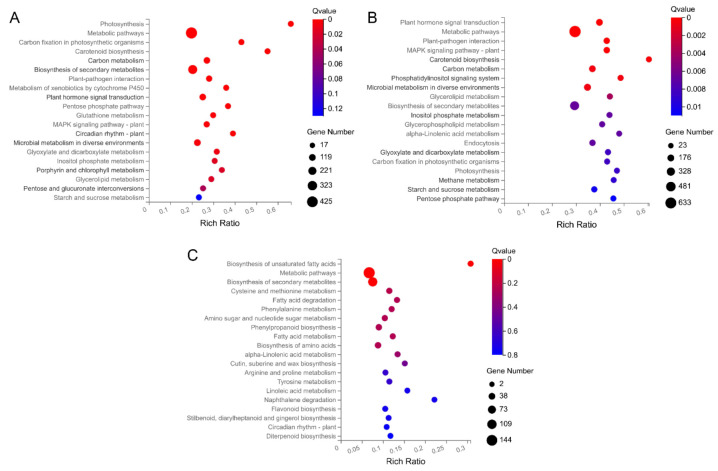
Bubble chart of KEGG pathway enrichment. (**A**) KEGG pathways enriched with upregulated DEGs from the MM_1-vs-MM_2 comparison. (**B**) KEGG pathways enriched with upregulated DEGs from the 20965_1-vs-20965_2 comparison. (**C**) KEGG pathways enriched with upregulated DEGs from the MM_2-vs-20965_2 comparison. The sizes of the bubbles indicate the number of genes enriched in the KEGG pathway. The color of the bubble represents the Qvalue. The rich factor is the ratio of differentially expressed gene numbers annotated in this pathway term to all gene numbers annotated in this pathway term. The greater is the rich factor, the greater is the degree of pathway enrichment.

**Figure 6 ijms-22-08128-f006:**
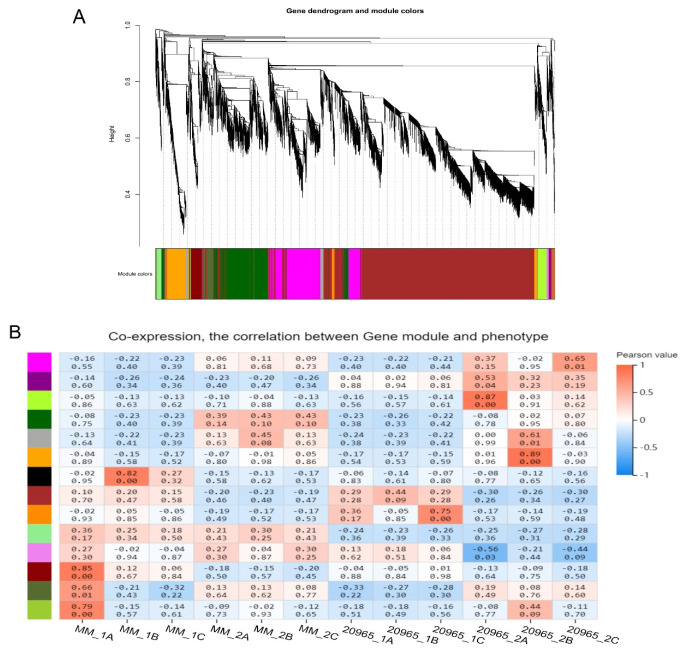
Analysis of expression patterns in tomato flower samples by WGCNA. (**A**) Gene clustering tree and module division. (**B**) Module-sample association. The abscissa represents the samples; the ordinate represents the modules. The upper value in each cell represents the correlation coefficient, and the lower value represents the *p* value.

**Figure 7 ijms-22-08128-f007:**
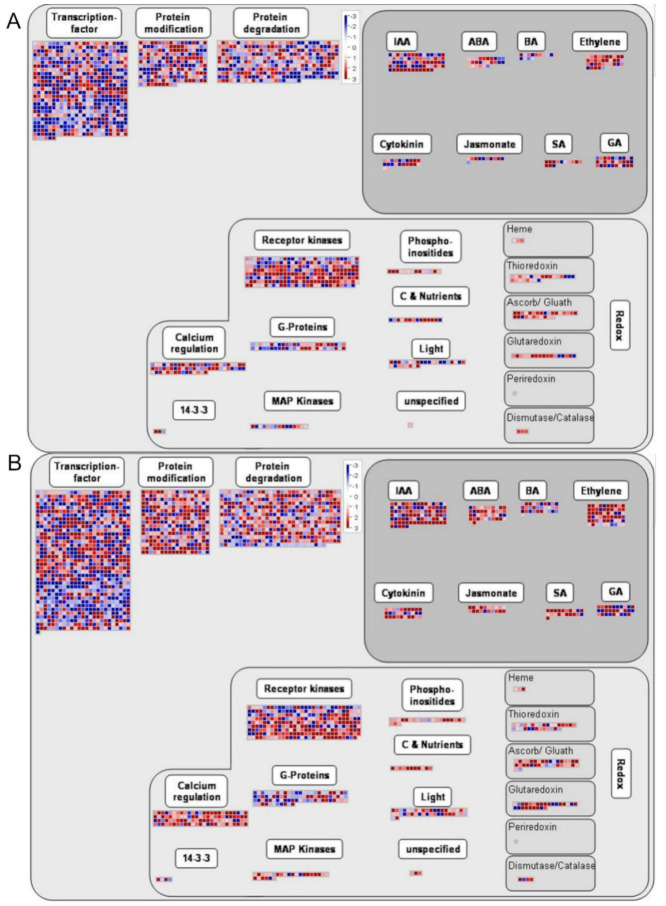
The clustering pattern of DEGs generated with the MapMan tool. (**A**) Overview of DEG regulation in MM1. vs-MM_2. (**B**) Overview of DEG regulation in 20965_1-vs-20965_2. (**C**) Cellular response of DEGs in MM_1-vs-MM_2. (**D**) Cellular response of DEGs in 20965_1-vs-20965_2. (**E**) Transcription of DEGs in MM_1-vs-MM_2. (**F**) Transcription of DEGs in 20965_1-vs-20965_2. Each square represents a separate gene. Red represents upregulation and blue represents downregulation. The color brightness represents the degree of difference (see scale).

**Figure 8 ijms-22-08128-f008:**
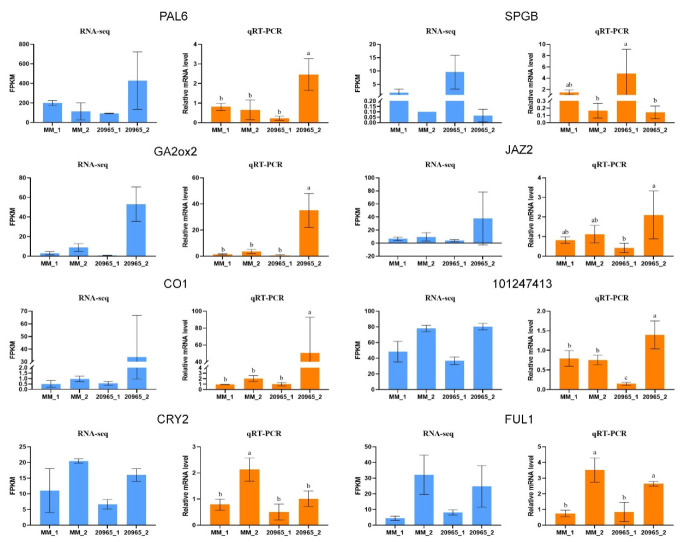
FPKM values obtained via RNA-seq and relative mRNA levels obtained via qRT-PCR for 8 DEGs. Three technical replicates were performed for each biological replicate of each sample. Error bars represent standard deviation. *p* < 0.05 means significant difference: a was significantly higher than b and c; b was significantly higher than c. There was no significant difference between the same letters.

**Figure 9 ijms-22-08128-f009:**
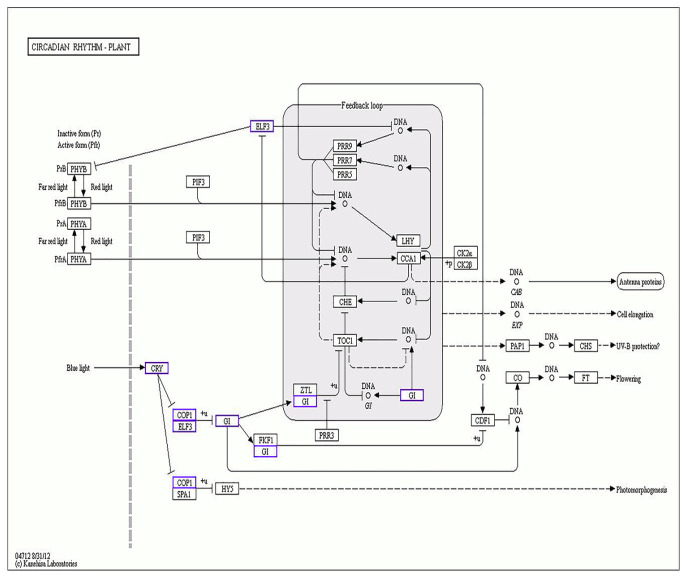
Network analysis of DEGs in the “circadian rhythm-plant” category. The small rectangular squares represent genes or proteins.The small dots represent chemical molecules. Straight lines and arrows represent activation. Dotted lines and arrows represent indirect effects. T-bars stand for inhibition. +u represents ubiquitination. +p represents phosphorylation. The purple squares represent DEGs enriched in this pathway in the comparison groups MM_1-vs-MM_2 and 20965_1-vs-20965_2.

**Table 1 ijms-22-08128-t001:** Statistics of the *Solanum lycopersicum* transcriptome database.

Sample	Raw Reads (M)	Clean Reads (M)	Clean Bases (Gb)	Q20 (%)	Q30 (%)	Clean Reads Ratio (%)	Total Mapping (%)	Uniquely Mapping (%)
MM_1A	45.57	44.22	6.63	97.9	94.07	97.04	94.44	92.71
MM_1B	47.33	45.43	6.81	97.89	94.08	95.98	93.54	91.97
MM_1C	45.57	43.92	6.59	97.97	94.34	96.37	93.85	92.19
MM_2A	45.57	43.95	6.59	97.87	94.03	96.44	92.9	91.05
MM_2B	45.57	44.32	6.65	97.91	94.13	97.24	93.12	91.54
MM_2C	45.57	44.41	6.66	97.68	93.5	97.44	94.57	93.01
20965_1A	45.57	43.95	6.59	97.27	92.66	96.43	93.62	91.94
20965_1B	47.33	45.02	6.75	97.45	93.17	95.13	93.14	91.47
20965_1C	45.57	44.35	6.65	97.3	92.75	97.31	94.21	92.55
20965_2A	45.57	44.23	6.63	97.39	92.99	97.06	92.86	90.6
20965_2B	45.57	44.34	6.65	97.28	92.68	97.28	93.73	92.03
20965_2C	45.57	44.03	6.60	97.33	92.82	96.6	93.42	91.39

“MM” represents Moneymaker; “MM_1” is the early stage of flower buds; “MM_1A”, “MM_1B”, and “MM_1C”are three samples of Moneymaker’s flower buds, respectively; “MM_2” is the flower extension period; “MM_2A”, “MM_2B”, and”MM_2C” are three samples of Moneymaker flower exhibition, respectively; “20965” represents the “20965” homozygous inbred line; “20965_1” is the early stage of 20965’s flower bud; “20965_1A”, “20965_1B”, and “20965_1C” are three samples of the 20965’s flower buds, respectively; “20965_2” is the flower extension period; “20965_2A”, “20965_2B”, and “20965_2C” are three samples of 20965’s flower exhibition, respectively.

**Table 2 ijms-22-08128-t002:** DEGs identified from different comparisons.

Comparison Group	Down	Up	Total
MM_1-vs-MM_2	2828	3152	5980
20965_1-vs-20965_2	3825	4954	8779
MM_1-vs-20965_1	565	214	779
MM_2-vs-20965_2	486	859	1345

According to the gene expression level of each sample, the number of upregulated and downregulated significant DEGs in each comparison group was obtained. The total is the sum of upregulated and downregulated DEGs.

**Table 3 ijms-22-08128-t003:** Tomato flowering time- and flower development-related genes in the MM_1-vs-MM_2 and 20965_1-vs-20965_2 comparisons.

Flowering Pathway	Gene ID	Gene Symbol	Log_2_ Fold-Change	
			MM_1-vs-MM_2	20965_1-vs-20965_2
Photoperiod	543547	COP1	0.30	1.15
Photoperiod	101259391	COP1-like	2.58	3.74
Photoperiod	101263075	COP1-interacting	0.53	1.78
Photoperiod	543596	CRY2	1.01	1.41
Photoperiod	778253	CO1	1.02	6.16
Photoperiod	100191137	CO	1.42	2.35
Photoperiod	101245415	CO3	1.29	1.73
Photoperiod	100134891	EZ1	0.59	1.15
Photoperiod	101245336	FT	1.95	2.26
Photoperiod	109119816	PELPK1-like	2.72	5.29
Photoperiod	101256688	ELF4-like	0.75	1.13
Photoperiod	101265040	ELF3-like	0.67	1.47
Photoperiod	101252363	BEE2	3.03	4.23
Photoperiod	101252729	bHLH63	2.06	4.05
Photoperiod	101265165	bHLH130	2.30	2.32
Photoperiod	101255722	WNK5	2.55	4.49
Photoperiod	101258346	GI-like	1.34	0.42
Vernalization	101244491	AGL42-like	1.26	3.29
Vernalization	101249083	AGL104	8.63	8.31
Gibberellin	544211	20ox-2	1.52	1.54
Gibberellin	100037505	GA2ox2	1.53	6.07
Gibberellin	100134889	GA2ox4	2.92	3.92
Gibberellin	543839	FUL1	2.90	1.76
Gibberellin	543887	FUL2	1.25	1.98
Gibberellin	101247719	LOC101247719	1.36	1.68
Gibberellin	101249868	SOC1-like	0.47	1.59
Gibberellin	101253625	GID1ac	1.59	0.60
Gibberellin	100736493	GID1b-1	2.04	0.32
Gibberellin	101258522	MYB21	7.64	11.52
Gibberellin	101265384	GAI	3.55	1.71
Age	100191138	AP2a	1.39	1.84
Age	101244632	AP2-like	−2.04	−3.56
Temperature	101245946	actin6	−1.00	−1.55
Temperature	101246885	actin7	−1.38	−1.72
Temperature	101266766	SVP	−0.66	−1.70
Temperature	101252303	PIF4	1.31	1.39
Autonomy	101246029	SFT	3.52	2.21
Autonomy	101252492	FRI	−0.84	−1.14
Autonomy	101251416	LOC101251416	6.15	5.36
	544020	PRK2	10.21	11.53
	544021	PRK1	11.03	12.13
	101243937	LOC101243937	6.84	6.88
	101245709	BOP3	−0.04	−1.99
	101246001	BOP2	−1.46	−1.27
	101247407	LOC101247407	1.18	1.89
	101246927	TAA1	4.30	6.68
	101252256	AIM1	1.11	2.38
	101251501	GPAT6	0.24	3.20
	100134911	JAZ2	0.58	3.38

## Data Availability

The raw sequencing data of this article are stored in the NCBI Sequence Read Archive under accession number GSE163914.

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
