# Peer review of "Transcriptome Analysis of Flower Development and Mining of Genes Related to Flowering Time in Tomato (Solanum lycopersicum)"

_ijms, 2021, doi:10.3390/ijms22158128_

Round 1
Reviewer 1 Report
The manuscript proposed by Wang et al. describes a transcriptomic analysis in tomato focused on flower development. The authors performed RNAseq experiments on two tomato varieties and two development stages of inflorescence.
Overall the data are well presented and executed. I have just a minor comment regarding the nomenclature used. The authors used "Control1" and "Control3" for the two varieties and "Treat1" and "Treat3" for the two developmental stages. This nomenclature is not very intuitive.
Usually, "Control" and "Treat" are used for stress conditions. I suggest to modify the nomenclature with more appropriate names.
Reviewer 2 Report
The authors compared the transcriptional profiles of a wild-type tomato and a recombinant imbred line (they do not specify whether it is an early or late flowering variety) to identify genes involved in flowering time regulation. The experimental setup, the number of replicates are appropriate. The bioinformatic analysis is well performed, the presentation of the results is mostly good but must be improved. The description of the methods should also be improved. Despite its scientific value (such works are important for the improvement of the functional annotation of the tomato genome), the manuscript is very difficult to read because of the poor English. There were countless of instances where I had to spend a considerable time to decipher what the authors wanted to tell. The manuscript could use a professional language editing service, in this form the MS is not suitable for publication. However, after correcting the text and the issues listed below, the MS should be considered for publication.
Major issues:
- Poor English and superfluous, long sentences that should be simplified. A professional language editing service could fix this problem. Beside this, there are countless of typos, missing spaces (i.e. before references, between sentences, after commas, etc).
- The IDs of genes are in some custom format (a number that was possibly assigned by the Hisat software) that are useless for the scientific community. Please rename or annotate the genes using the commonly used SolycXXgXXXXXX format (in figures and tables and in the supplementary materials as well). For the novel genes, the genomic co-ordinates should be provided.
- Problems with figures:
The bottom of Fig 3 is missing.
Fig 4 is not informative since it does not tell anything about the enrichment of the shown categories compared to the background. The x-axis title says "Number" instead of "Number of genes", and the y-axis says "Level2" instead of "GO terms". It is not specified which samples were compared in this figure. Actually, the corrected p-value (q-value) for the enrichment for the terms would be more appropriate to show in such a figure. I prefer the bubble chart representation of GO or KEGG term enrichments that the authors chose in case of the other figures (i.e. Fig 5).
The bottom and the right part of the Fig 5 are missing. A major problem with this figure is that the definition of the Rich ratio (x-axis) is not given in the figure legend (i.e. "The Rich factor is the ratio of differentially expressed gene numbers annotated in this pathway term to all gene numbers annotated in this pathway term. The greater the Rich factor, the greater the degree of pathway enrichment").
Fig 6 is duplicated. Also, I think most of the modules identified could be collapsed into five different modules if we consider the samples (three replicates each). The reason for the separation is the difference between the replicates. With different settings these modules could be collapsed. Also, it is not clear to me what the p-values represent here. What was the statistical test for?
Fig 7 is very difficult to understand. I get that I should compare panel A to B, C to D, and E to F, but what to compare? I guess every dot is one gene belonging to the given category and the color means that it is up- or downregulated between the specified samples. Shell I compare the number of genes in a category in A and B panels? Or their colors? An explanation of this figure in the figure legend would be very helpful. Also, there is no description of this analysis in the Methods part.
In Fig 8 the error bars of the RNA-seq expressions are missing despite that there are three biological replicates. Also, it is not mentioned what the error bars represent in the case of RT-qPCR experiments. Are they Standard Deviation or Standard Errors? Are they technical or biological replicates? Based on the Methods part, I guess the authors performed three technical replicates for each biological replicates of each samples (or just three technical replicates for one sample?). It should be specified in the figure legend and in the Methods part as well.
Fig9: More detailed description of the figure in legends needed. What are the colored boxes (DEGs)? - Specify the phenotype of the inbred line: early or late flowering, how the homozygosy was determined (with reference).
Minor issues:
- Line 26: Delete "the life history of"
- Line 27: Delete: "and the completion of the first step of sexual reproduction"
- Line 32: Change "The regulation of flowering time is achieved via a complex network system and is mainly affected by internal genetic factors and external environmental changes" to "Flowering time is regulated by internal and external factors"
- Line 34: Rephrase sentence "When..."
- Line 57: Delete sentence "One core gene of the vernalization pathway in A. thaliana is green LOCUS C (FLC), which protects against early FLOWERING by inhibiting FT expression" because the previous sentence says the same.
- Line 60: Change "regulating the positive expression of FT" to "induces the expresion of FT"
- Line 64: Rephrase "the degree of development and genetic background of indivudual tomato plants play" like "the developmental stage and genotype play"
- Line 72: Perhaps you wanted to write "sympodial" instead of "symbiotic"
- Line 237: Fix "。" to ". "
- Line 293: Change "upregulation of DEG expression was greater than the downregulation of DEG expression" to "There were more upregulated genes than downregulated genes"
- Line 300: To my knowledge, there is no "vinyl signaling", perhaps yo wanted to write "ethylene signaling"
- Line 325: Delete "extremely" and "the process of"
- Line 328: Change "maintain life continuity" to "survive"
- Line 321: The figure legend should be more detailed, specify the number of replicates, the definition of error bars. A statistical test should be performed to determine if the differences between the means of the specified samples are significant (for the RT-qPCR studies only). Correct "FRKM" to "FPKM".
- Line 333-336: Very complicated sentences, rephrase like "According to the DEG analysis, there is a bigger difference between the developmental stages within a genotype than between genotypes at the same developmental stage", or something.
- Line 349: Delete "can be regarded as"
- Line 357: Delete sentence "For tomato flowering regulation, photoperiod is a very important method."
- Line 358: Change "Photoreceptors (PHYA, PHYB, CRY, etc.) are plant components " to "PHYA, PHYB, CRY, etc, are photoreceptors"
- Line 372: Delete "activities"
- Line 374: Change italics to regular because these are proteins. By the way, check the MS for gene, protein, and mutant name formats.
- Line 377: Change "its similar proteins" to "its homologues" and from "GI-like protein genes" delete "protein"
- Line 379: Fix "eraches" to "reaches"
- Line 381: From "rhythm cycle" delete "cycle"
- Line 397: Change "CO and similar proteins" to "CO and similar genes"
- Line 398: Change "extension" to "expansion" and "CO is" to "CO"
- Line 409: Fix "tomatot" to "tomato"
- Line 464: Fix "SPTULA" to "SPATULA"
- Line 511: "fragmented with ion interruption". I do not know about this method. Is there any reference or kit name that utilizes this method?
- Line 514: "by terminal pairing". I have no idea wat this means.
- Line 518: "150 bp". Was it paired-end or single-end? Stranded or unstranded (I guess the latter one).
- Line 521: Delete "information"
- Line 528: "by comparing the filtered data". I guess it should be "by alignment of the filtered sequences"
- Line 534: "error expression rate" should be corrected to "false discovery rate"
- Line 535: Instead of explaining what Venn analysis is, a reference to the tool used should be provided.
- Line 537-538: The sentence can be deleted.
- Line 545: Fix "will be" to "was"
- Line 550: From "biochemical metabolic pathways" delete "biochemical" and "pathways"
- WGCNA analysis description should be provided with more details.
- Line 553: Delete "to complete the WGCNA procedure."
- Line 556: Delete "based on the highest value" (what does it mean, anyway?) and "to display the variations in genetic correlation between each module."
- Line 562: Which kit was used? How many biological and technical replicates were performed? A statistical test should be performed to determine if the differences between the means of the specified samples are significant (for the RT-qPCR studies only). Specify the criteria for statistical significance.
Reviewer 3 Report
Wang et al. analyzed transcriptomes of early flower buds and flower extension period using two tomato cultivars differing in flowering times to select DEGs related to the flowering time and flower development. As transition from the vegetative to reproductive growth represents important part of the plant development with respect to the prospective fruit yield, complex understanding of this process is supposed to be implemented consecutively in breeding programs. The submitted work has a potential to contribute to this knowledge. Nevertheless, there is space for improvement of the manuscript quality, as suggested below:
- In Introduction, SP5G gene is reported as a positive regulator and TMF gene as a negative regulator of flowering but both mutants display early flowering phenotypes (l. 86-90). Please clarify.
- Tomato cultivars differing in their flowering times were selected for analyses (l. 102-103), but no relevant details are stated in the text (in Results or in Material and Methods), and results are not presented and discussed with respect to this fact. I strongly recommend to explain properly why these two cultivars were selected and to stress the benefits of this selection.
- There is stated in the text (l. 183-184) that “different flower development period in the two materials” were compared (Figure 5C). Based on the Figure legend, KEGG pathways enriched with DEGs from the Control3-vs-Treat3 comparison are presented, thus, in my opinion, the same flower development periods (flower extension period) in two cultivars (“materials”) were compared. Please clarify.
- Specify how candidate genes related to the flowering time and flower development were selected; 40 candidates are presented in the Chapter 2.2 based on the DEGs evaluation (l. 145-146), analysis of 75 candidates is described in the Chapter 2.4.
- I was rather confused by the description of transcription patterns of flowering-related genes presented in the Chapter 2.4 and Table 3. For example, COP1-related genes are presented as up-regulated in the text (l. 206-207) but according to the Table 3 data, some of them are bellow log2-fold change > 1 in Control1-vs-Control3 comparison with this value selected as a cut-off for DEGs (l. 134). Please add relevant discussion.
- Figure 7 is difficult to read in the printed version.
- Unify the References style (compare, e.g., Ref. 4 and Ref. 26).
Round 2
Reviewer 3 Report
I appreciate changes done by Wang et al. and/or their explanations.
During the manuscript processing, I recommend to pay attention to many typos; e.g., two dots at the end of the sentence (l. 294, 486); redundant dot (l. 303); imperfectly clipped Figures 4 and 5; missing „more“ (l. 283); the sentence „We used mRNA fragments as templates to reverse transcrib cDNA (l. 565-566) does not make sense neither with „transcribe“; log2-fold change>1 and Q value <0.05 in Results (l. 133) but >1 and <0.05 in Methods (l. 582); style of References not unified completely (e.g., Ref. 54 – full length journal name).
